# The Potential Effect of Annealing Mesostructured Titanium Dioxide Electrode in a Closed Box Furnace on the Concentration of Lead (II) Iodide Solution Required for Optimal Performance of Mesoscopic Perovskite Solar Cells

**Muhammad Talha Masood** [1,2,*], **Amna Safdar** [1], **Muhammad Aftab Akram** [1], **Sofia Javed** [1] and **Syeda Qudsia** [2]

[1]  Department of Materials Engineering, School of Chemical & Materials Engineering, National University of Science & Technology (NUST), H-12 Sector, Islamabad 44000, Pakistan; amna.safdar@scme.nust.edu.pk (A.S.); aftabakram@scme.nust.edu.pk (M.A.A.); sofia.javed@scme.nust.edu.pk (S.J.)

[2]  Laboratory of Molecular Science & Engineering, Faculty of Science & Engineering (FNT), Åbo Akademi University, Henrikinkatu 2, FI-20500 Turku, Finland; syeda.qudsia@abo.fi

\*  Correspondence: talha.masood@scme.nust.edu.pk; Tel.: +92-300-5151099

**Abstract:** Highly reproducible mesoscopic perovskite solar cells (PSCs) can be fabricated using two-step sequential deposition of organo-lead halide (perovskite) active layer. However, differences in the processing conditions of individual layers which are subsequently assembled to construct the ultimate device can result in variations in the solar cell performance. For instance, here we report trends in the device performance as a function of $PbI_2$ solution concentration, where the compact and mesoporous $TiO_2$ layers were annealed in a closed box furnace (instead of doing it in open air). We observed that the devices prepared using 1.2 M $PbI_2$ solution concentration performed better than those prepared from 0.8 M and 1 M $PbI_2$ solutions. Generally, the researchers use the hot plate in an open-air environment or use a special hot plate where a continuous flow of air is ensured while annealing $TiO_2$ electron selective layers (ESL) for perovskite solar cells. In this case, the highest possible device efficiencies are achieved using 1 M concentration of $PbI_2$ solution. Although the influence of $PbI_2$ solution concentration has been previously studied in detail, here our prime focus is to briefly comment on slight differences in the device performance trends which we observed in comparison to the previously reported results, where $TiO_2$ layers were calcined in open air.

**Keywords:** Perovskite solar cells (PSCs); Power conversion efficiency (PCE); Lead (II) Iodide ($PbI_2$); Methylammonium Iodide (MAI); Methylammonium Lead Iodide perovskite ($MAPbI_3$); $PbI_2$-to-$MAPbI_3$ conversion; $PbI_2$/$TiO_2$ interface; perovskite/$TiO_2$ interface



## 1. Introduction

Solar energy is freely and abundantly available form of energy which can be effectively used for electricity generation using solar cells. The solar cells based on crystalline silicon are well commercialized and known to be reliable because they are technologically mature in comparison to other types of photovoltaic devices. However, highly energy intensive and expensive processing methods required for manufacturing silicon wafers have pushed the scientific community to look for some other alternatives such as perovskite solar cells (PSCs). The high-power conversion efficiencies (PCEs) of PSCs are known to be very comparable to those of silicon-based solar cells. The fabrication of PSCs is also known to be very cost-effective due to their solution-processability. Earlier research in this area depended on one-step deposition of perovskite active layer, in which a mixed precursor single solution was spin-coated on top of the substrate [1]. The synthesis of the active layer using a one-step approach would result in its highly non-uniform coverage on top of the substrate, thus large number of parallel shunts would deteriorate the device performance and its reproducibility. The introduction of two-step sequential deposition of the perovskite

active layer by Burschka et al. [2] significantly solved this problem. In this method, the lead halide solution was spin-coated, followed by dropping the Methylammonium halide solution on top of the rotating substrate [3,4] or immersing the lead halide coated substrate into the Methylammonium halide solution [2]. There were still issues with this method, because to bring their PCEs beyond 15 to 16% required major modifications, such as interdiffusion strategy. This is generally because of the incomplete conversion of $PbI_2$ to perovskite upon reaction with MAI, which results in a considerable amount of unreacted $PbI_2$ in the deeper regions of perovskite active layer [5]. To further enhance the device efficiencies, one-step deposition was again introduced, but this time it involved dropping the Chlorobenzene [6,7] or Toluene [8] as anti-solvents on top of the rotating substrate comprising of single precursor solution. The anti-solvent approach potentially produces high-efficiency devices, but still suffers from reproducibility issues. This is because this route is highly delicate and sensitive to handling and deposition conditions [9]. Thus, the two-step deposition technique is relatively more tolerant to such issues and therefore highly reproducible, with some compromise on the device efficiencies owing to the presence of residual $PbI_2$ in the perovskite active layer.

On the other hand, this unreacted $PbI_2$ is also known to be beneficial for the passivation of interfacial defects at the perovskite/metal oxide electrode interface [10]. However, there is always some critical amount of unconverted $PbI_2$ in the perovskite active layer beyond which device performance starts to deteriorate. The amount of unconverted $PbI_2$ in the perovskite can be controlled either by controlling the $PbI_2$ solution concentration while keeping the concentration of MAI solution constant, or vice versa [11]. The optimal device performance with the two-step approach can be achieved by keeping the $PbI_2$ and MAI solution concentrations as 1 M and 0.063 M, respectively. Tuning the MAI solution concentration significantly influences the crystal size of the perovskite active layer, thus bulk recombination can be somehow minimized by reducing the number of grain boundaries. On the other hand, the $PbI_2$ concentration influences the morphology of the perovskite capping layer on top of the mesostructured substrate [11,12].

Several research works performed in the past few years investigated the influence of $PbI_2$ and MAI solution concentrations. Here, we also present the effect of $PbI_2$ solution concentration on the device performance, though we will discuss some differences in trends we observed as a function of $PbI_2$ concentration in comparison to the previously reported results which suggested that 1 M $PbI_2$ solution is most suitable for the optimal device performance. We used three different concentrations of the $PbI_2$ solution, i.e., 0.8 M, 1 M and 1.2 M, respectively. The solar cells prepared using 1.2 M $PbI_2$ solution were found to perform unexpectedly better than the devices prepared using 1 M $PbI_2$ solution. However, it is a well-known fact that 1 M concentration of $PbI_2$ solution is best for the optimal performance of PSCs. Our XRD results also revealed that the residual $PbI_2$ content in the perovskite films prepared using 1.2 M $PbI_2$ solution is significantly higher. Thus, one should expect that the lower perovskite content in 1.2 M sample should ideally not be able to give enough photogenerated electricity. Surprisingly, the $V_{OC}$ values of the devices based on 1.2 M $PbI_2$ concentration were found to be significantly higher than those prepared using 1 M concentration of $PbI_2$ solution. We suspect that the mesoporous $TiO_2$ used as a scaffold in this work contains very high concentration of oxygen vacancies because we used a closed box furnace for its calcination with no continuous air flow. This has been previously demonstrated by Ho et al. [13], that a continuous oxygen-rich air flow is required during the calcination of $TiO_2$ films to minimize oxygen defects which can be the ultimate source of surface recombination at the $TiO_2$/Perovskite interface [14]. Oxygen defects at the surface of $TiO_2$ thin films can also be reduced using UV-ozone treatment [15]. The presence of oxygen vacancies tends to disrupt the entire stoichiometric balance on both sides of the $TiO_2$/Perovskite interface. This results in partial breaking up of Pb-I bonds of the perovskite close to the interface. Thus, there are defects in the perovskite as well as in the $TiO_2$. The defects in perovskite tend to capture the photogenerated electrons within the perovskite, which can potentially reduce the $J_{SC}$. On the other hand, interfacial trap states

due to the oxygen vacancies in $TiO_2$ captures the injected electrons which subsequently recombine with the holes in VB of the perovskite (because of electron back transfer) which ultimately reduces the $V_{OC}$ [14]. Since we expect that our (compact and mesoporous) $TiO_2$ comprises of higher density of oxygen vacancies, their effective passivation requires much larger quantity of unreacted (or residual) $PbI_2$ at the perovskite/$TiO_2$ interface. This is possible by using higher $PbI_2$ solution concentration (such as 1.2 M) than previously optimized concentration (i.e., 1 M).

## 2. Materials and Methods

### 2.1. Materials

Fluorine-doped Tin Oxide coated glass substrates (TCO22-15) were purchased from Solaronix (Aubonne, Switzerland). $TiCl_4$ (>99%) was bought from Fluka (Seelze, Germany). Pluronic F127 block co-polymers, tetrahydrofuran (THF, >99%), *N,N*-dimethylformamide (DMF, 99.8%), 2-propanol (99.5%), chlorobenzene (99.8%), acetonitrile (99.8%), and Lithium bis(trifluoromethane)sulfonimide (Li-TFSI), were purchased from Sigma Aldrich (St. Louis, MO, USA). The $PbI_2$ (99.99%) and Methylammonium Iodide, MAI (>98%) were acquired from TCI Europe (Zwijndrecht, Belgium) and spiro-OMeTAD from Feiming Chemicals Ltd (Shenzhen, China). The cobalt (III) tri[bis-(trifluromethane]sulfonamide) (FK209 Cobalt (III) salt) and 30 NR-D $TiO_2$ paste were obtained from Greatcell (Queanbeyan, Australia). Ethanol (EtOH, >99.5%) was bought from ALTIA Plc, Helsinki, Finland.

### 2.2. Preparation of Compact and Mesoporous TiO₂ Films

The compact $TiO_2$ (c-$TiO_2$) dip coating solution was prepared as previously reported by Masood et al. [16]. For this, a $TiCl_4$/EtOH stock solution was prepared by dropwise addition of 18.97 g of $TiCl_4$ into 23.04 g of EtOH while stirring in an ice bath. Another solution comprising of 0.0152 g of F127 block co-polymer, 12.34 g of EtOH, 0.21 g of deionized water and 1.717 g of THF was prepared. Thereafter, 2.50 g of the $TiCl_4$/EtOH stock solution was added dropwise into the second solution and kept for stirring for at least 30 min before dip coating the FTO coated glass substrates.

The FTO coated glass substrates were cut into 4 cm × 2 cm pieces. A 1.5 cm × 2 cm area was etched from one side of the FTO substrates using Zn powder and 4 M aqueous solution of HCl. The substrates were sonicated in 2% aqueous Halmanex III solution, de-ionized water, Acetone and 2-propanol for 10 min each. After drying the substrates by blowing dry Nitrogen, they were further subjected to plasma treatment for 5 min before dip-coating in the c-$TiO_2$ solution. For dip-coating, the withdrawal speed of 85 mm/min was used without allowing any dwell time in the solution. The c-$TiO_2$ substrates were then dried on the hot plate at 125 °C followed by direct calcination at 500 °C for 30 min in the closed box furnace. To study the perovskite filled mesostructured $TiO_2$ (by XRD, UV-Vis and FE-SEM), the unetched FTO coated glass substrates were used. The subsequent mesoporous $TiO_2$ layer was deposited as a scaffold, thus preparing the mp-$TiO_2$/c-$TiO_2$/FTO/glass substrates. For this purpose, the suspension of $TiO_2$ nanoparticles was prepared by dissolving 30 NR-D $TiO_2$ paste (from Greatcell) in Ethanol (EtOH) to maintain the concentration of 0.15 g/mL. After applying plasma treatment on top of c-$TiO_2$/FTO/glass substrates, the mesoporous $TiO_2$ solution was spin-coated using 4000 rpm spin speed for 10 s. The spin acceleration was set to 2000 rpm/s. The films were then dried on the hot plate at 120 °C for 10 min before calcination at 450 °C for 30 min by following the previously reported heating ramp [16,17].

### 2.3. Deposition of the Perovskite Active Layer

Next, 0.8 M, 1 M and 1.2 M $PbI_2$ solutions were prepared in anhydrous DMF while stirring at 100 °C on hot plate. The solutions were kept at 100 °C throughout the experiment to prevent precipitation of $PbI_2$ crystals within the solutions. The $PbI_2$ solutions were spin-coated on the mesostructured $TiO_2$ substrates with a spin speed of 6000 rpm for 30 s using 6100 rpm/s acceleration rate. The $PbI_2$-coated substrates were dried on a hot plate at 100 °C for 30 min. Upon subsequent cooling to room temperature, the substrates were dipped in

10 mg/mL solution of MAI dissolved in anhydrous isopropanol followed by quick rinsing in isopropanol to remove excess MAI. The substrates were again dried at 100 °C for 30 min on the hot plate.

### 2.4. Characterization

The Grazing incidence X-ray diffraction (GI-XRD) was performed on all the perovskite infiltrated mesostructured $TiO_2$ (i.e., mp-$TiO_2$/c-$TiO_2$/unetched-FTO/glass) substrates. This was carried out to verify the differences in the relative amount of perovskite with respect to the unreacted $PbI_2$ as a function of $PbI_2$ solution concentration. This was achieved using Bruker AXS D8 Discover instrument (Billerica, MA, USA). The Bragg's angle scan was performed between 10° to 20° using 0.04° step size at grazing incidence angle of 1°. The UV-visible (UV-vis) absorption scan was also performed on these samples between 300 to 900 nm wavelength range using Perkin Elmer Lambda 900 UV-vis near infrared spectrometer (Waltham, MA, USA). The slit size was kept at 2 mm and the measurements were performed in the presence of certified reflectance standards. The UV-Vis measurements were also performed on $PbI_2$ infiltrated mesostructured $TiO_2$ (mp-$TiO_2$/glass) substrates as a reference.

To assess the morphology of $PbI_2$ and perovskite capping layers on top of the mesostructured $TiO_2$ layer, the field emission scanning electron microscopy (LEO, Oberkochen, Germany) was performed. For this purpose, the 50 k times magnification, 2.70 kV electron high tension and an aperture size of 10 μm were used.

### 2.5. Device Fabrication

The lab-scale devices were prepared using different $PbI_2$ concentrations by following our previously reported protocol [16]. Figure 1 schematically shows the structure of our typical device.

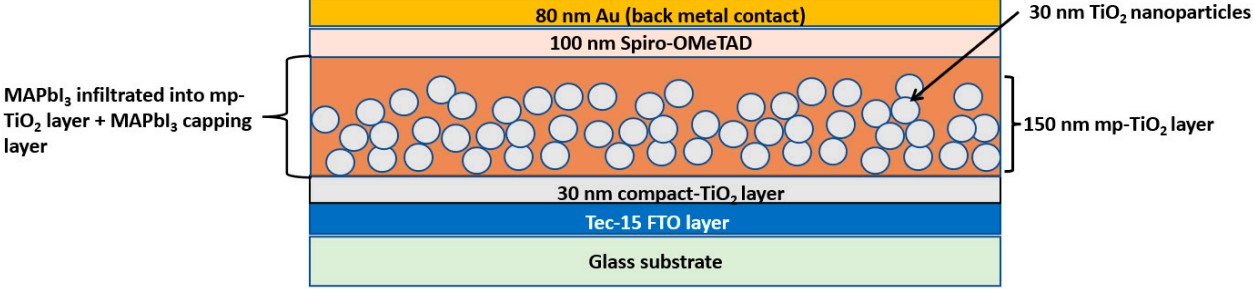

**Figure 1.** The typical device structure and the layer thicknesses used in this study.

The etching of FTO, and the deposition of compact and mesoporous $TiO_2$ layers has already been explained in Section 2.2. Three to four devices for each $PbI_2$ concentration were fabricated. After depositing the perovskite active layer, as mentioned in Section 2.3, the spiro-OMeTAD and 4-tertbutyilpyridine were dissolved in chlorobenzene. On the other hand, the solutions of Lithium bis(trifluoromethylsulfonyl)imide and FK209 Co(III) were prepared by dissolving them in acetonitrile. These solutions were then added to the solution of Spiro-MeTAD. This was carried out to maintain the molar ratio of $1.0{:}0.5{:}2.5 \times 10^{-2}{:}3.3{:}131.5{:}7.2$ (spiro-MeTAD: Li-TFSI: FK209 Co (III): 4-tertbutylpyridine: chlorobenzene: acetonitrile). The final solution was then spin-coated on top of the perovskite active layer using a spin speed of 4000 rpm for 30 s with 4000 rpm/s spin acceleration. The circular gold back metal contacts of area 0.283 $cm^2$ were deposited on top of the Sprio-OMeTAD layer. This was achieved by evaporating gold at an evaporation rate of about 0.01 nm/s while keeping the vacuum pressure of about $2 \times 10^{-5}$ mbar.

### 2.6. Device Characterization

The J-V curves were measured in ambient conditions under illumination using a simulated sunlight at AM 1.5 with 100 mW/cm$^2$ intensity (Oriel Class ABB solar simulator, 150 W, 2″ × 2″). A circular aperture with the size of about 0.126 cm$^2$ was used to mask the devices. The J-V scan was performed using a 2636 Series Source Meter (Keithley instruments, Cleveland, OH, USA)—The scan was started from −0.3 V to 1.1 V in forward sweep followed by 1.1 V to −0.3 V in reverse sweep using a scan speed of about 10 mV/S [17]. For each PbI$_2$ concentration at least 3 to 4 devices were characterized to determine the photovoltaic parameters.

## 3. Results

### 3.1. Determining the Perovskite Content Using GI-XRD

We started by analyzing the influence of PbI$_2$ solution concentration on PbI$_2$-to-MAPbI$_3$ conversion efficiency in PbI$_2$ infiltrated mesostructured TiO$_2$ substrates (when dipped in 0.063 M MAI solution, as discussed in Section 2.3). This was carried out by performing GI-XRD using 1° incident angle upon each sample after it was converted to the perovskite. The diffractograms from GI-XRD were then normalized to further determine the relative amounts of remnant PbI$_2$ and the perovskite. The samples in the subsequent text are named according to the concentration of PbI$_2$ solutions. The samples without conversion to the perovskite were named as 0.8 M PbI$_2$, 1.0 M PbI$_2$ and 1.2 M PbI$_2$. The samples after conversion to the perovskite were named as 0.8 M MAPbI$_3$, 1.0 M MAPbI$_3$ and 1.2 M MAPbI$_3$.

The normalized GI-XRD diffractograms are shown in Figure 2, where all the perovskite samples contain both the perovskite (MAPbI$_3$) and the unreacted residual PbI$_2$. This is evident from PbI$_2$ (001) and MAPbI$_3$ (110) reflections at 12.6° and 14.1° angles, respectively.

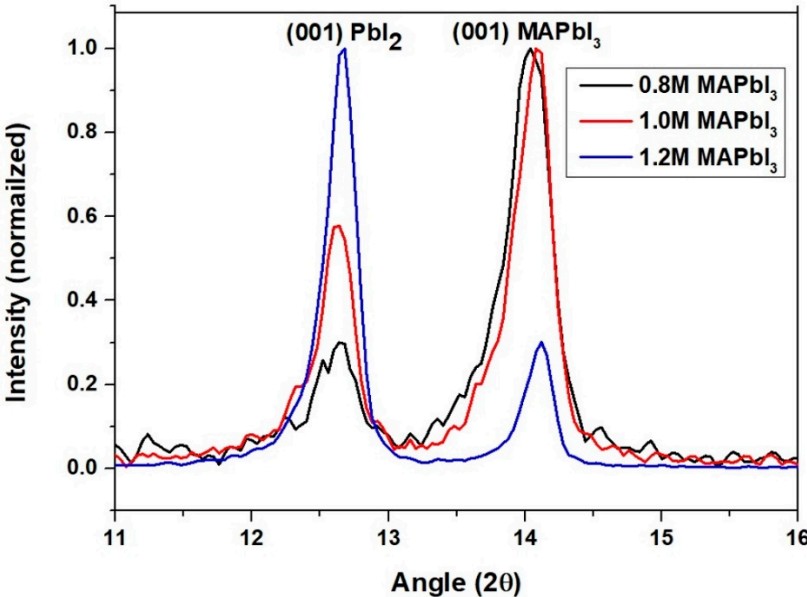

**Figure 2.** Normalized GI-XRD diffractograms of the perovskite films prepared using PbI$_2$ solutions of three different concentrations.

The unreacted PbI$_2$ is left in the deeper regions of all the perovskite samples because MAI is not able to reach those areas to react and form MAPbI$_3$ [17]. Using areas under each reflection, we calculated the percentage of perovskite (MAPbI$_3$) in each sample based on different PbI$_2$ solution concentrations. Areas under each reflection were calculated by integration and then by using the following formula, and the percentage of perovskite in each sample was calculated:

$$\% \, Perovskite = \frac{Area \; under \; MAPbI_3(110) \; peak}{Area \; under \; MAPbI_3(110) \; peak + Area \; under \; PbI_2(001) peak} \times 100\% \quad (1)$$

The percentage perovskite can be used as a measure of $PbI_2$-to-$MAPbI_3$ conversion efficiency. A higher percentage of perovskite ($MAPbI_3$) peak is a sign of better conversion efficiency and vice versa. Table 1 shows the percentage perovskite values calculated from Figure 2.

**Table 1.** Percentage of perovskite ($MAPbI_3$) content in the samples as a function of $PbI_2$ solution concentration

.

| $PbI_2$ Concentration | % Perovskite |
|---|---|
| 0.8 M | 74.48 |
| 1.0 M | 62.89 |
| 1.2 M | 23.62 |

The trend in these values demonstrates that that the $PbI_2$-to-$MAPbI_3$ conversion is suppressed upon increasing the concentration of $PbI_2$ solution, which is absolutely expected. This is because, moving towards increased concentration of $PbI_2$ solution means increased and dense crystallization of $PbI_2$ within the mesostructured layer. Therefore, $PbI_2$ in the deeper regions of mesostructured $TiO_2$ is not accessible for Methylammonium Iodide (MAI) (when dipped in the MAI solution). An elaborate discussion regarding the $PbI_2$-to-$MAPbI_3$ conversion can be found in one of our previous works [17].

### 3.2. Influence of $PbI_2$ on the Absorbance before and after Conversion to $MAPbI_3$

The samples were further analyzed using UV-Vis spectroscopy and the resulting spectra are shown in Figure 3.

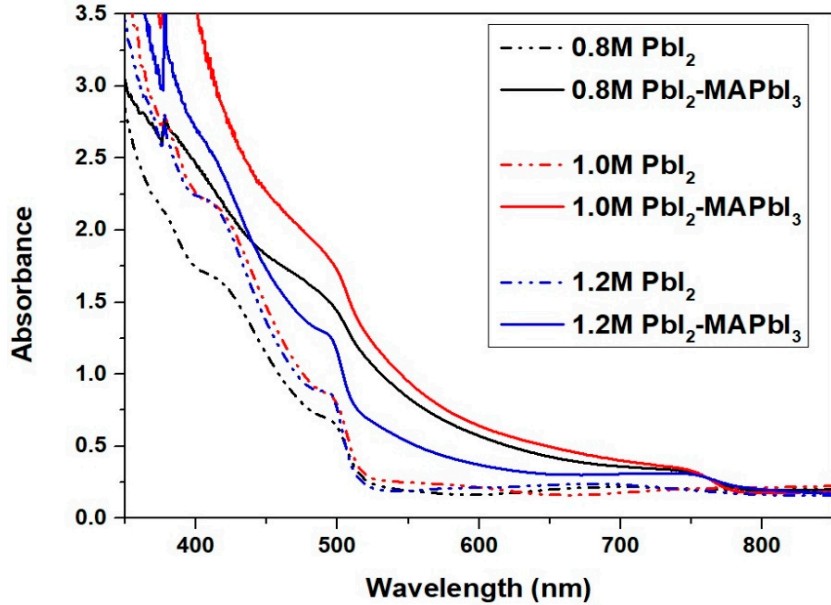

**Figure 3.** UV-Vis absorption spectra for $PbI_2$ coated mesostructured $TiO_2$ substrates using different $PbI_2$ solution concentration (shown as dotted lines) and corresponding perovskite films (shown as solid lines).

The dotted lines show the spectra corresponding to the mesostructured $TiO_2$ substrates infiltrated with $PbI_2$ by spin-coating solutions of different $PbI_2$ concentrations. The same figure also shows the spectra of $MAPbI_3$ corresponding to the samples with three

different PbI$_2$ concentrations. The spectra corresponding to the PbI$_2$ loaded films reveals that increasing PbI$_2$ concentration from 0.8 to 1 M results in a significant increase in the absorbance below 500 nm wavelength. However, further increase in the concentration to 1.2 M does not result in any noticeable change. This is probably because increasing the PbI$_2$ concentration results in increased loading of mesostructured TiO$_2$ with PbI$_2$, which is densely crystallized within the porous channels and in the form of capping layer. Going beyond 1 M concentration results in the removal of excess PbI$_2$ during spinning process. In other words, there is a limit, beyond which further loading of mesostructured TiO$_2$ films with PbI$_2$ is not possible, hence we do not see any further increase in absorbance when moving from 1 to 1.2 M PbI$_2$ loaded mesostructured TiO$_2$. Contrary to this, using lower concentrations of PbI$_2$ (such as 0.8 M) probably results in sparse crystallization of PbI$_2$ within the porous channels of mesostructured TiO$_2$.

The perovskite (MAPbI$_3$) corresponding to the 1 M PbI$_2$ concentration shows highest absorbance over the entire wavelength range while lowest absorbance (above 440 nm wavelength) is achieved for the perovskite prepared from 1.2 M PbI$_2$ solution. Our results agree with Wang et al. [11], who also performed similar measurements on ZnO doped TiO$_2$ nanostructured films loaded with MAPbI$_3$ prepared from different PbI$_2$ concentrations.

In general, these results suggest that 1 M PbI$_2$ concentration is most suitable for optimal conversion to MAPbI$_3$ using 0.063 M MAI solution. Although the highest PbI$_2$-to-MAPbI$_3$ conversion percentage is achieved for 0.8 M MAPbI$_3$ sample (based on GI-XRD results), its absorbance is still lower in comparison to the 1 M MAPbI$_3$. This indicates that the overall amount of perovskite in the 0.8 M MAPbI$_3$ is still lower in comparison to the perovskite prepared using 1 M PbI$_2$ concentration. The absorbance of 1.2 M MAPbI$_3$ is even inferior to 0.8 M MAPbI$_3$ and 1 M MAPbI$_3$ samples within the visible range. This is due to its poorest PbI$_2$-to-MAPbI$_3$ conversion efficiency. Thus, it can be stated for sure that the overall amount of perovskite in 1 M MAPbI$_3$ sample is still higher than the perovskite in 0.8 M and 1.2 M samples.

### 3.3. Morphology of the Capping Layer before and after Conversion to MAPbI$_3$

Figure 4 shows the SEM images of PbI$_2$ coated mesostructured TiO$_2$ substrates prepared using three different PbI$_2$ concentrations and their corresponding MAPbI$_3$ perovskite samples (after conversion).

It is evident from Figure 4a that the complete coverage of PbI$_2$ capping layer could not be achieved in 0.8 M PbI$_2$ sample. This is because this PbI$_2$ concentration is barely sufficient to fill in the pores while its capping layer can partially cover the mesoporous TiO$_2$ layer. The underlying TiO$_2$ nanoparticles left uncovered by PbI$_2$ capping layer are also visible as white spots in Figure 4a. For 1 M and 1.2 M PbI$_2$ samples (shown in Figure 4b,c, respectively), the PbI$_2$ capping layer completely covers the underlying mesostructured TiO$_2$.

The SEM images of corresponding 0.8 M, 1 M and 1.2 M MAPbI$_3$ samples are shown in Figure 4d–f, respectively. The pinholes in the perovskite capping layer are clearly visible for 0.8 M and 1 M MAPbI$_3$ samples. These pinholes may allow the infiltration of subsequent hole selective layer (HSL) down to the underlying mesostructured TiO$_2$ during its deposition process and enable the formation of parallel shunts to deteriorate the device performance. However, the complete coverage of the perovskite capping layer is achieved using 1.2 M concentration of PbI$_2$ (i.e., for 1.2 M MAPbI$_3$ sample shown in Figure 4f). This could possibly have significant contribution to the trends in the device performance as a function of PbI$_2$ concentration (typically the $V_{OC}$ values).

Based on GI-XRD, UV-Vis and SEM results, increased PbI$_2$ concentration results in increased loading of mesostructured TiO$_2$. In other words, filling up of the empty spaces within the mesostructured TiO$_2$ by PbI$_2$ is proportional to the PbI$_2$ solution concentration. PbI$_2$ may also densely fill in the voids between TiO$_2$ nanoparticles upon using even higher concentration of PbI$_2$ solution such as 1.2 M.

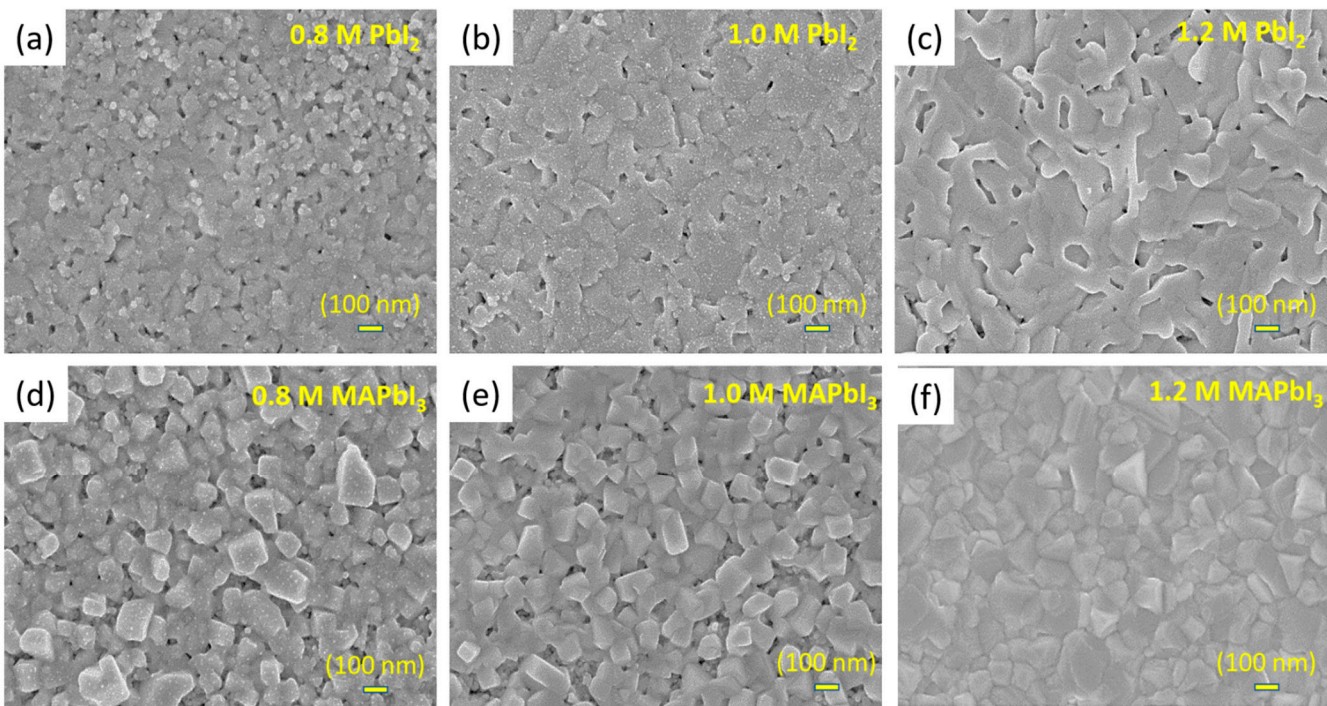

**Figure 4.** (**a**–**c**). SEM images of $PbI_2$ capping layers on top of mp-$TiO_2$ using different $PbI_2$ solution concentration; (**d**–**f**) SEM images of corresponding perovskite capping layers after conversion.

Increased loading of $PbI_2$ within the mesostructured $TiO_2$ can be beneficial in terms of achieving higher perovskite content, provided that the $PbI_2$ is not densely crystallized within the porous channels. This is because $PbI_2$ in deeper regions should be accessible for effective reaction with MAI to form $MAPbI_3$. If $PbI_2$ is densely crystallized within the pores, MAI will not be able to reach $PbI_2$ lying in the deeper regions of mesostructured $TiO_2$ for complete reaction. It is also possible that the conversion is even restricted to the capping layer region with little conversion near the pore openings. This seems to be more obvious for 1.2 M $MAPbI_3$ sample in Figure 2 (GI-XRD diffractogram). On the other hand, using lower $PbI_2$ concentration (such as 0.8 M) may result in sparse crystallization of $PbI_2$ within the pores. Therefore, $PbI_2$ crystallized in deeper region of the porous channels are more accessible for MAI to react. Therefore, the conversion of $PbI_2$-to-$MAPbI_3$ is more efficient for lower $PbI_2$ concentration, and vice versa.

*3.4. The Influence of PbI_2 Concentration on the Device Performance*

The devices were prepared using a perovskite absorber based on three different $PbI_2$ concentrations. Figure 5 shows the light J-V curves under forward and reverse sweep for the devices performing close to average power conversion efficiencies (PCEs). Three to four devices were analyzed for each $PbI_2$ concentration to tabulate the mean and standard deviation for each photovoltaic parameter shown in Table 2.

The table shows decreasing trends in the fill factor with increase in $PbI_2$ concentration. This decreasing trend is more obvious upon increasing $PbI_2$ concentration from 0.8 to 1 M. This means that an increased amount of unreacted $PbI_2$ in the perovskite light absorber tends to reduce the photoconversion upon illumination as a function of $PbI_2$ concentration. Conversely, lower content of unreacted $PbI_2$ tends to positively influence the fill factor, and vice versa. The enhancement in $V_{OC}$ due to increased concentration of $PbI_2$ solution can be attributed to the increased coverage of perovskite capping layer. We already mentioned this while discussing the SEM images. Despite drastic increase in the unreacted $PbI_2$ content with increased $PbI_2$ solution concentration (especially in the case of the 1.2 M $MAPbI_3$ sample), the $V_{OC}$ tends to increase, which positively influences the PCE. Upon increasing the $PbI_2$ concentration from 0.8 to 1 M, an average $V_{OC}$ of our solar cells in reverse sweep

increases by 6.15%. It was very surprising that, upon increasing $PbI_2$ concentration from 1 to 1.2 M, the average $V_{OC}$ in reverse sweep increases by 10.5%. This sudden increase in $V_{OC}$, despite limited amount of light harvesting $MAPbI_3$ phase in the 1.2 M $MAPbI_3$ sample can be attributed to the passivating effect of unreacted $PbI_2$ which potentially passivates all kinds of interfacial defects at the perovskite/$TiO_2$ interface [18,19]. This might also include physical or chemical passivation of oxygen vacancies acting as mid bandgap recombination centers at the perovskite/$TiO_2$ interface. This raises a question that why there is a significant increase in $V_{OC}$ despite the presence of high content of unreacted $PbI_2$ in 1.2 M sample (i.e., around 76%). This relative amount of unreacted $PbI_2$ should ideally create a potential barrier for electron injection into $TiO_2$ and significantly reduce the $V_{OC}$ and FF values. FF is somehow reduced, but why is $V_{OC}$ enhanced?

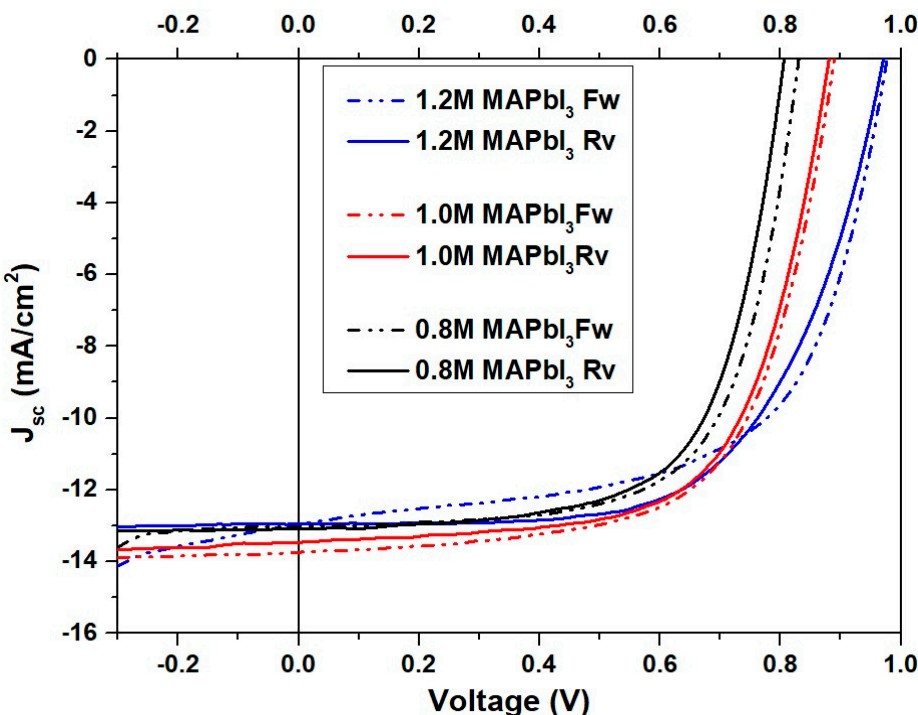

**Figure 5.** J−V curves under illumination using perovskite active layer prepared using three different $PbI_2$ concentrations; 0.8 M, 1.0 M and 1.2 M.

**Table 2.** Mean and standard deviations of photovoltaic parameters measured in (a) forward and (b) reverse sweep for the PSCs prepared from different $PbI_2$ concentration.

| (a) Forwards Sweep | | | | |
|---|---|---|---|---|
| $PbI_2$ Concentration | $J_{SC}$ (mA/cm$^2$) | $V_{OC}$ (V) | FF | PCE (%) |
| 0.8 M | 13.115 ± 0.374 | 0.839 ± 0.013 | 0.673 ± 0.025 | 7.425 ± 0.485 |
| 1.0 M | 13.850 ± 0.365 | 0.891 ± 0.017 | 0.620 ± 0.024 | 7.655 ± 0.342 |
| 1.2 M | 13.507 ± 0.939 | 0.975 ± 0.004 | 0.617 ± 0.006 | 8.120 ± 0.607 |
| (b) Reverse Sweep | | | | |
| $PbI_2$ Concentration | $J_{SC}$ (mA/cm$^2$) | $V_{OC}$ (V) | FF | PCE (%) |
| 0.8 M | 13.155 ± 0.392 | 0.829 ± 0.021 | 0.693 ± 0.022 | 7.579 ± 0.532 |
| 1.0 M | 13.395 ± 0.556 | 0.880 ± 0.022 | 0.635 ± 0.017 | 7.470 ± 0.321 |
| 1.2 M | 13.407 ± 0.800 | 0.972 ± 0.008 | 0.620 ± 0.010 | 8.130 ± 0.638 |

## 4. Discussion

We speculate that there is a higher concentration of oxygen vacancies on the surface of $TiO_2$ nanoparticles because we performed the calcination of mesoporous $TiO_2$ layer in

a closed box furnace instead of in the open air or in an environment with continuous air flow. Calcination of TiO$_2$ in open air or under continuous air flow can significantly reduce the oxygen vacancies and thus minimize the stochiometric imbalance on the surface of TiO$_2$ nanoparticles. The surface of TiO$_2$ (nanoparticles) directly establishes its interface with the subsequently deposited layer, such as the perovskite (in this case). Ho et al. analyzed the performance of PSCs as a function of calcination in oxygen-rich and oxygen-deficient environments, respectively. They confirmed that annealing TiO$_2$ films under oxygen-deficient environment significantly gives rise to higher concentration of oxygen vacancies on TiO$_2$ surface and deteriorates the device performance [13]. Saliba et al. strongly recommend the use of a hot plate with continuous air flow for calcination instead of using closed box furnace to ensure high efficiency of PSCs [8]. It is also recommended to perform UV-ozone or TiCl$_4$ treatments of the TiO$_2$ scaffold prior to the deposition of perovskite to remove oxygen defects at the perovskite/TiO$_2$ interface [15,18]. In such conditions, the best device performance can possibly be achieved using 1 M PbI$_2$ concentration when using 2-step sequential deposition of perovskite on top of mesoporous TiO$_2$. However, in our case, we only had the facility to perform calcination of compact and mesoporous TiO$_2$ in the closed box furnace.

We propose that annealing TiO$_2$ samples in closed box furnace could have resulted in higher content of oxygen vacancies than if the annealing was performed on hot plate in open air. This is a well-known fact; that cooler air is denser than the hotter air. Upon using a hot plate in the open air, only the substrates are at annealing temperature (i.e., around 450 °C), while the air on top of the substrates is cooler and is at room temperature. Thus, when TiO$_2$ coated substrates are annealed in open air, their surfaces are expected to have continuous exposure to the oxygen because the cooler air on top of the substrates is denser. This can potentially reduce the amount of oxygen vacancies in the surface region of TiO$_2$ films (during annealing process). On the other hand, when the closed box furnace is used for calcination, both the samples and the air inside the furnace chamber are at a higher temperature (i.e., at 450 °C). The hot air inside the closed box furnace is lighter and probably does not make substantial contact on top of the TiO$_2$ surface. Thus, the samples' surfaces do not have sufficient exposure to oxygen during annealing in a closed box furnace, and we would ideally expect higher concentration of oxygen vacancies close to the TiO$_2$ film surface.

There could be another explanation; that while annealing the samples in closed box furnace, there is no continuous exchange of air between the calcination chamber (where the samples are placed) and the region outside the furnace. In contrast, the substrates being annealed on the hot plate in open air are expected to be exposed to an infinite amount of air which continuously supplies oxygen molecules to the sample surface, which can potentially play an important role in minimizing the oxygen vacancies. Since our samples were calcined in a closed box furnace, we expect more oxygen defects at the perovskite/TiO$_2$ interface than if the calcination was performed on hot plate in open air. Thus, our sample devices probably require more unreacted PbI$_2$ for defects passivation than what is achieved using 1 M PbI$_2$ concentration. Thus, using 1.2 M concentration of PbI$_2$ solution, gives significantly large amount of unreacted PbI$_2$ which establishes a large interfacial contact with TiO$_2$ nanoparticles. This restricts the electron injection into TiO$_2$ within few areas where perovskite is directly able to establish its interface with TiO$_2$. This also limits the oxygen vacancies coming in the path of photogenerated electrons being injected into TiO$_2$. In other words, the amount of residual PbI$_2$ required for passivation of oxygen vacancies is proportional to the oxygen vacancy concentration in TiO$_2$.

The passivation of oxygen defects by residual PbI$_2$ in the mesostructured TiO$_2$ is schematically explained in Figure 6, where oxygen vacancies are marked as active sites (in red). These defects will be coming in the path of injected electrons and behave as mid-gap states. These states tend to trap many injected electrons, which subsequently recombine with the holes in the perovskite valance band. This phenomenon is also known as electron back-transfer [14,20] and sometimes called the surface recombination [21].

Excess or unreacted $PbI_2$ in the embedded perovskite is able make such oxygen vacancies inactive or passivated, which are marked in white color.

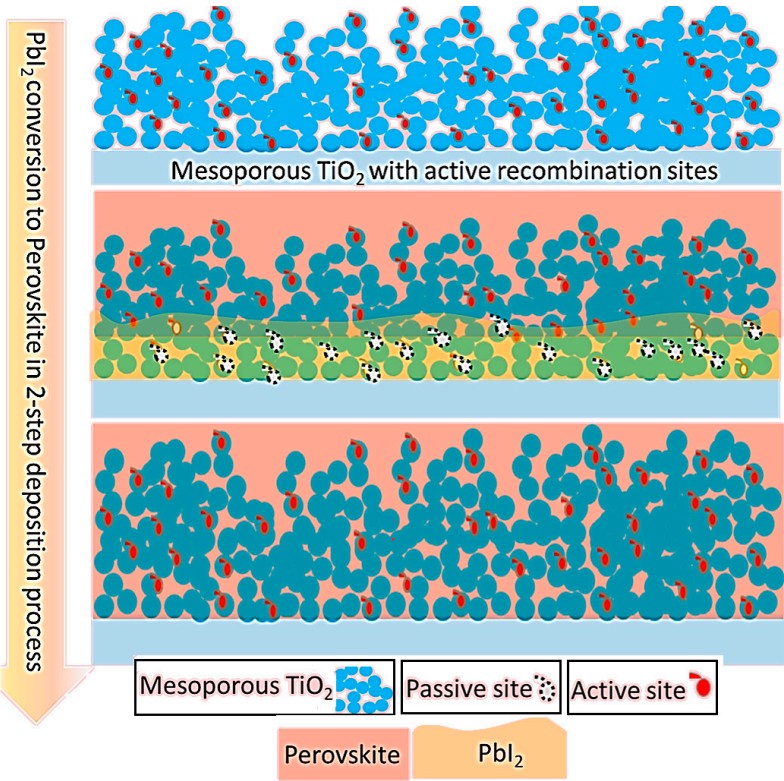

**Figure 6.** The presence of unreacted $PbI_2$ passivates the active oxygen defects. Thus, efficient conversion of $PbI_2$ to the perovskite is not ideal to achieve efficient PSCs.

If $PbI_2$-to-$MAPbI_3$ conversion is more complete, then there is large interfacial contact area between the perovskite and $TiO_2$, which means that more mid-gap active defects are also expected to be encountered by injected electrons. If the density of such active defects is very high, then it is useful to deactivate them by passivation using relatively higher concentration of $PbI_2$ solution such as 1.2 M. We believe that 1.2 M concentration of $PbI_2$ solution (instead of 1 M concentration) helped us to minimize the active defect sites to increase the $V_{OC}$. It is also important to keep in mind that very high content of unreacted $PbI_2$ is also not desirable because it may not leave any region where the perovskite may establish its interface with $TiO_2$. In that case, the remnant $PbI_2$, which establishes its interface with $TiO_2$, may only act as a potential barrier against electron injection.

Since the conversion initiates from the top and proceeds towards the bottom, the perovskite/$TiO_2$ interface seems to be more concentrated close to the topmost portion of the mesostructured $TiO_2$ layer while the deeper region mostly comprises of $PbI_2$/$TiO_2$ interfaces due to the incomplete conversion.

It is also possible that the oxygen defects are more concentrated in the deeper regions of the mesostructured $TiO_2$ layer due to the restricted transportation of oxygen in that region during calcination in the closed box furnace. When using the closed box furnace with no continuous air flow, the fixed amount of oxygen in this environment is probably just able to minimize the defects close to the surface of mesostructured $TiO_2$ film (i.e., the area which is most accessible for oxygen). Thus, in such circumstances, the efficient conversion of $PbI_2$ to the perovskite is not ideal for fabricating efficient devices. Rather, a large portion of unreacted $PbI_2$ is important, which can be ensured by using the $PbI_2$ solutions with concentrations above 1 M; although this concentration is well-known to be optimal for PSCs when using two-step sequential deposition of perovskite photoactive layer.

## 5. Conclusions

The influence of $PbI_2$ solution concentration on the performance of PSCs (when using two-step sequential deposition of perovskite) has been frequently reported in the past, where 1 M $PbI_2$ concentration is well-known to be suitable for optimal device performance. In this work, we found that the devices prepared from 1.2 M concentration of $PbI_2$ solution perform better than those prepared from 1 M $PbI_2$ solution, owing to the enhancement in $V_{OC}$. Increased $PbI_2$ solution concentration resulted in increased remnant $PbI_2$ content in the perovskite ($MAPbI_3$), which plays a crucial role in the enhancement of $V_{OC}$s. Lower unreacted $PbI_2$ content (upon using lower $PbI_2$ concentration such as 0.8 M) was observed to positively influence the fill factor. However, significant increase in the $V_{OC}$, when using $PbI_2$ concentration beyond its optimal limit of 1 M was a bit surprising. We speculate that apart from enhanced coverage of perovskite capping layer for 1.2 M sample, higher unreacted $PbI_2$ content plays an important role in interfacial defects passivation. In our case, the mesostructured $TiO_2$ was calcined in a closed environment without any air flow, which might have resulted in higher density of oxygen vacancies at the perovskite/$TiO_2$ interface than if we were able to perform calcination in an open-air environment or at least under continuous air flow. Therefore, it is most likely that relatively more $PbI_2$ is required to passivate them which is possible by using higher $PbI_2$ solution concentration than 1 M. Thus, by increasing the residual $PbI_2$ content between $TiO_2$ and perovskite, the interfacial area for the current extraction is significantly reduced. This potentially reduces a large number of oxygen vacancies coming in the path of injected electrons. Thus, the photogenerated electrons in the perovskite can only be injected into $TiO_2$ via restricted regions, where perovskite forms direct interface with $TiO_2$. By restricting the interfacial contact area between the perovskite and $TiO_2$, defect density coming along the path of injected electrons is significantly reduced. Reduced number of trap-states can be quickly saturated in the beginning of device operation under illumination. Thus, the only trajectory left for the electrons would be to smoothly be injected into $TiO_2$ via limited regions of perovskite/$TiO_2$ interface and be extracted into the external circuit via conductive FTO.

**Author Contributions:** Conceptualization, M.T.M. and A.S.; methodology, M.T.M. and S.Q.; validation, M.A.A., S.J. and A.S.; formal analysis, M.T.M., A.S. and S.Q.; investigation, M.T.M. and S.Q.; resources, M.T.M. and S.Q.; data curation, M.T.M. and S.Q.; writing—original draft preparation, M.T.M. and A.S.; writing—review and editing, all authors.; visualization, M.T.M., A.S. and S.Q.; supervision, M.T.M.; project administration, M.T.M. and S.Q.; All authors have read and agreed to the published version of the manuscript.

**Funding:** This research was funded by FDP program sponsored by National University of Science & Technology (NUST), Pakistan and the Academy of Finland, project number 308307.

**Acknowledgments:** The authors are thankful to Jouko Peltonen, Jan-Henrik Smått and Ronald Österbacka for providing administrative and technical support. The authors specially thank Ronald Österbacka, who is currently the head of the physics department in Åbo Akademi University, Finland, since he provided with full access to the equipment required for device fabrication and analysis.

**Conflicts of Interest:** The authors declare no conflict of interest. The funders had no role in the design of the study; in the collection, analyses, or interpretation of data; in the writing of the manuscript, or in the decision to publish the results.

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
