# Peer review of "The Potential Effect of Annealing Mesostructured Titanium Dioxide Electrode in a Closed Box Furnace on the Concentration of Lead (II) Iodide Solution Required for Optimal Performance of Mesoscopic Perovskite Solar Cells"

_crystals, doi:10.3390/cryst12060833_

Round 1
Reviewer 1 Report
The authors investigated the impact of PbI2 solution concentrations on device performance with compact and mesoporous TiO2 layers annealed in a closed box furnace. The study is interesting, but some details are missing. Some comments are as follows.
- The term “silicon” should not be capitalized.
- Section 2: the thickness of layers is missing. I would suggest the authors add a structure/device schematic.
- Section 2.5: the 0.01 nm/s evaporation rate of gold is too small, and I will doubt the accuracy of the detector, considering the gold atom is about 0.16 nm.
- The formula on page 6 is missing.
- The reference list should be fixed.
Author Response
Reviewer’s comments:
The authors investigated the impact of PbI2 solution concentrations on device performance with compact and mesoporous TiO2 layers annealed in a closed box furnace. The study is interesting, but some details are missing. Some comments are as follows.
- The term “silicon” should not be capitalized.
- Section 2: the thickness of layers is missing. I would suggest the authors add a structure/device schematic.
- Section 2.5: the 0.01 nm/s evaporation rate of gold is too small, and I will doubt the accuracy of the detector, considering the gold atom is about 0.16 nm.
- The formula on page 6 is missing.
- The reference list should be fixed.
Response to Reviewer 1:
Special thanks to the reviewer for useful feedback and comments. We appreciate your earnest and prompt response, and hope that this revision will meet with approval. Our response to your concerns is listed as follows:
- This issue has been addressed.
- The figure has now been added which shows the device structure and thickness of each layer
- During this research, we tried to keep the minimum possible evaporation rate of Gold to avoid its penetration through the spiro (i.e., the hole transport layer) down to the perovskite capping layer. We used the same evaporation rate in our earlier studies which can be found in the following links.
- https://pubs.acs.org/doi/10.1021/acsami.7b02868
- https://www.sciencedirect.com/science/article/abs/pii/S0040609019304377
- The formula on page 6 was earlier mentioned in the main manuscript however it disappeared when the manuscript was uploaded. Unfortunately this happened with both the word and pdf versions of the manuscript. We have now fixed this problem. Thank you for notifying this.
- The reference list is now fixed.

Reviewer 2 Report
The authors in their work have studied the effect of PbI2 solution concentration in the perovskite formation and its effect on the performance of a solar cell. The difference of this study with respect to others is that here compact and mesoporous TiO2 layers were annealed in a closed box furnace instead of doing it in open air.
The materials used and the methodology followed to develop the study are well described. In general, the manuscript presents a good bibliographic review and justification of the work. the results and, above all, the discussion and analysis of the results is very good. This made it feasible to come to good and effective conclusions. However, there are two questions for the authors which I detail below.
1.- The integral equation used to determine the area under the peaks in the X-ray diffraction analysis is not observed in the manuscript.
2.- In section 2.2 (Preparation of compact and mesoporous TiO2 films) it is indicated that a substrate was etched from one side of the FTO using Zn powder and 4 M aqueous solution of HCl, ¿would you explain for what purpose you do this? and another issue neither the results nor their discussion present any analysis of the effect of having attacked the substrate with Zn and HCl, ¿what can you comment on this?
Author Response
Reviewer’s comments:
The authors in their work have studied the effect of PbI2 solution concentration in the perovskite formation and its effect on the performance of a solar cell. The difference of this study with respect to others is that here compact and mesoporous TiO2 layers were annealed in a closed box furnace instead of doing it in open air.
The materials used and the methodology followed to develop the study are well described. In general, the manuscript presents a good bibliographic review and justification of the work. the results and, above all, the discussion and analysis of the results is very good. This made it feasible to come to good and effective conclusions. However, there are two questions for the authors which I detail below.
- The integral equation used to determine the area under the peaks in the X-ray diffraction analysis is not observed in the manuscript.
- In section 2.2 (Preparation of compact and mesoporous TiO2films) it is indicated that a substrate was etched from one side of the FTO using Zn powder and 4 M aqueous solution of HCl, ¿would you explain for what purpose you do this? and another issue neither the results nor their discussion present any analysis of the effect of having attacked the substrate with Zn and HCl, ¿what can you comment on this?
Response to the reviewer’s comments:
Special thanks to the reviewer for useful feedback and comments. We appreciate your earnest and prompt response, and hope that this revision will meet with approval. Our response to your concerns is listed as follows:
- This was mentioned in the main manuscript however it somehow disappeared when the manuscript was uploaded. This happened with both the word and pdf versions of the manuscript. Now we have fixed this problem. Thanks for notifying this.
- The FTO coated glass was etched from one side. One end of the gold metal electrodes should lie on top of the spiro-OMeTAD-based hole transport layer, while the other end of the gold contact should be in the etched region. If FTO is not removed from this region, the gold contacts will be in direct contact with FTO causing direct shunts. If this region is not properly etched, i.e., when some FTO still remains, there is a risk of contact-related issues in the devices. The concept is illustrated in the following diagram (please find it in the attachment).
The reviewer also pointed out that “neither the results nor their discussion present any analysis of the effect of having attacked the substrate with Zn and HCl, ¿what can you comment on this?”
It is important to point out, that for perovskite solar cells and organic solar cells, the transparent conductive metal oxides layers (on top of glass) such as FTO (Fluorine doped Tin oxide) or ITO (Indium doped Tin oxide) are always etched from one side the commercially available substrates. This is due to the “above-mentioned” reason. The effect of etching was not the focus of this study so we have not discussed it anywhere. The role of Zn and 4M HCl solution is just to etch away the FTO from the desired region of the FTO coated glass.

Reviewer 3 Report
The authors of this Manuscript describe the effect of the variable concentration of the reacting PbI2 solutions on the properties and performance of the final perovskite material. The English language is of a sufficient level for publication. The Conclusions are almost well supported by experimental data, even though they can be improved. Overall, I suggest the acceptance of this work only after a Minor Revision.
Here below are summarized my main concerns:
- Page 6: the formula used to calculate the percentage of perovskite in each sample is not visible.
- The authors speculate on the surface properties of the synthetized materials. Nevertheless, not enough results were acquired in order to sustain these claims. XPS studies must be performed on the proposed materials.
Author Response
Reviewer’s comments:
The authors of this Manuscript describe the effect of the variable concentration of the reacting PbI2 solutions on the properties and performance of the final perovskite material. The English language is of a sufficient level for publication. The Conclusions are almost well supported by experimental data, even though they can be improved. Overall, I suggest the acceptance of this work only after a Minor Revision.
Here below are summarized my main concerns:
- Page 6: the formula used to calculate the percentage of perovskite in each sample is not visible.
- The authors speculate on the surface properties of the synthetized materials. Nevertheless, not enough results were acquired in order to sustain these claims. XPS studies must be performed on the proposed materials.
Response to the reviewer’s concerns:
Special thanks to the reviewer for useful feedback and comments. We appreciate your earnest and prompt response, and hope that this revision will meet with approval. Our response to your concerns is listed as follows:
- The formula on page 6 was clearly written in the main manuscript however it disappeared when the manuscript was uploaded on mdpi website. Unfortunately, this happened with both the word and pdf versions of the manuscript. We have now fixed this problem.
- Indeed, this is a very important point. We earlier had this thing in our mind to go for XPS, to compare the difference between the concentration of oxygen vacancies in our samples (which were annealed in the closed box oven) with the samples annealed in open air. Unfortunately, we do not have access to XPS or AES for such analysis. Due to this restriction, we had to speculate based on literature for instance Ho et al (ref 13 in the manuscript). They did XPS measurements on TiO2 films which were annealed in oxygen rich and oxygen deficient environment respectively. Using XPS, they clearly observed that the strength of Ti3+2p3/2 signal reduces with increased annealing time in oxygen rich environment. This is the indication of reduction in oxygen vacancies and stochiometric imbalance near TiO2 surface (when prolonged annealing is performed under oxygen rich environment).
In this work, we have “proposed” based on our speculation that annealing TiO2 in closed environment could result in relative higher content of oxygen vacancies than if the annealing was performed in open air. When using hot plate, only the substrates are at annealing temperature (i.e., around 450oC) while the air on top is at room temperature. On the other hand, when the closed box oven is used, both the air and the substrates are at 450oC. This is a well-known fact that the cooler air is denser than the hotter air. Thus, when the TiO2 coated substrates are annealed in open air, their surfaces are expected to get continuous exposure to the oxygen because the air is denser. This could reduce the amount of oxygen vacancies on the surface of TiO2 film during annealing process. On the other hand, hot air in the closed box furnace is lighter and probably does not make substantial contact on top of the TiO2 film surface.
Secondly, by annealing the samples in closed box oven, there is no continuous exchange of air in the oven with that which is outside the oven. On the other hand, the substrates being annealed on the hot plate in open air is exposed to infinite amount of air which continuous supplies oxygen molecules to the sample surface. We think that this could also probably play some role in minimizing the oxygen defects on TiO2 surface.
We have now added this explanation on page 13 of the manuscript in the discussion section.

Round 2
Reviewer 1 Report
The manuscript has been improved and it is better presented, so I recommend it be published.
This manuscript is a resubmission of an earlier submission. The following is a list of the peer review reports and author responses from that submission.